# Indicators of the ozone recovery for selected sites in the Northern Hemisphere mid-latitudes derived from various total column ozone datasets (1980–2020)

Janusz Krzyścin[1]

[1]Institute of Geophysics, Polish Academy of Sciences, Warsaw, 01-452, Poland

*Correspondence to*: Janusz Krzyścin (jkrzys@igf.edu.pl)

**Abstract**.  We propose a method to examine the current status of the ozone recovery attributed to ozone-depleting substances (ODS) changes in the stratosphere. The total column ozone ($TCO_3$) datasets used are based on the ground-based (by the Dobson and/or Brewer spectrophotometer) measurements, satellite observations (from the Solar Backscatter Ultraviolet, SBUV, and Ozone Mapping and Profiler Suite, OMPS, instrument), and output of reanalyzes (Multi-Sensor Reanalysis version 2, MSR2, and Modern-Era Retrospective Analysis for Research and Applications, version 2, MERRA2). The $TCO_3$ time series are calculated for selected sites in the NH mid-latitudes (35–60° N), which are station locations with long-term $TCO_3$ observations archived at the World Ozone and Ultraviolet Radiation Data Centre (WOUDC).  The $TCO_3$ monthly means (1980–2020) are averaged over the April–September period to obtain $TCO_3$ time series for the warm sub-period of the year.  Two types of the averaged $TCO_3$ time series are considered:  the original one and non-proxy time series with removed natural variability by a standard multiple regression model.  $TCO_3$ time series were smoothed by the locally weighted scatterplot smoother and the super smoother. The smoothed $TCO_3$ values in 1980, 1988, 1997, and 2020 are used to build Ozone Recovery Indices (ORIs) in 2020. These are key years in the equivalent effective stratospheric chlorine (EESC) time series for the period 1980–2020, i.e., the stratosphere was only slightly contaminated by ODS in 1980, 1988 is the year in which the EESC value is equal to its value at the end (2020), and the EESC maximum was in 1997 in mid-latitude stratosphere. The first proposed ORI, $ORI_1$, is the normalized difference between the $TCO_3$ values in 2020 and 1988. The second one, $ORI_2$, is the percentage of the recovered $TCO_3$ in 2020 since the ODS maximum. Following these definitions, the corresponding reference ranges from –0.5 % to 1 % for $ORI_1$ and from 40 % to 60 % for $ORI_2$ are obtained analyzing a set of possible EESC time series simulated via the Goddard automailer. The ozone recovery phases are classified comparing the current ORI values and their uncertainty ranges (by the bootstrapping) with these reference ranges. In the analyzed $TCO_3$ time series, for specific combinations of datasets, data types, and the smoother used, we find faster (for $ORI_1$ or $ORI_2$ above the reference range), slower (for $ORI_1$ or $ORI_2$ below the reference range) recovery in 2020 than that inferred from the EESC change, and a continuation of $TCO_3$ decline after the EESC peak ($ORI_2 < 0$ %). Strong signal of the slower $TCO_3$ recovery is found in Toronto, Hohenpeissenberg, Hradec Kralove, and Belsk. A continuation of ozone decline after the turnaround in ODS concentration is found both in the original and non-proxy time series from WOUDC (Toronto), SBUV&OMPS (Toronto, Arosa, Hohenpeissenberg, Uccle, Hradec Kralove, and Belsk), and MERRA2 data (Arosa, Hohenpeissenberg, Hradec Kralove, and Belsk).

## 1 Introduction

Unexpected low total column ozone ($TCO_3$) values observed in the early 1980s over Antarctica alarmed both scientists and public because of anticipated increase of ultraviolet radiation (UVR) reaching the Earth's surface (Chubachi, 1984; Farman et al., 1985; Solomon et al., 1986). Widespread threats of thinning of the stratospheric ozone layer and corresponding danger for the Earth's environment led to signing the Montreal Protocol (MP) in 1987 to phaseout of the man-made ozone depleting substances (ODS). Overturning of the ODS concentration in the stratosphere (from large increase since the early 1980s to slight decrease beginning in the mid-1990s) was an evident sign of the success of MP and its subsequent amendments. The

ODS turnaround in the stratosphere was observed around the middle 1990s in the mid-latitudes and at the beginning of the 2000s in Antarctica. This also triggered numerous studies to reveal the corresponding change point also in $TCO_3$ trends to support impact of the MP and its later amendments on the ozone layer (e.g. Reinsel et. al., 2005; Mäder et al., 2007; Harris et al., 2008; Weber et al., 2018).

Trends in ozone were usually calculated using multiple linear regression (MLR) with a number of proxies to eliminate $TCO_3$ variations related to dynamical oscillations in the atmosphere, the solar 11-year activity, and volcanic eruptions. The anthropogenic component of the trend term was usually modelled as independent (disjoint) or dependent (joint) two lines drawn for the periods of increasing and decreasing ODS concentration in the stratosphere (e.g. Weber et al., 2018). There were only a few papers using a non-linear smoothed trend pattern based on dynamical linear model (Laine et al., 2014; Maillard Barras et al., 2022), Fourier series (Bozhkova et al., 2019), and other smoothers, e.g. locally weighted scatterplot smoothing (LOWESS) by Krzyścin and Rajewska-Więch (2016) and wavelets by Delage et al. (2022).

Coldewey-Egbers et al. (2022) calculated latitude/longitude linear $TCO_3$ trends since 1995 over the entire globe based on the satellite data record. The trends in the NH mid-latitudes were mostly insignificant. In some isolated regions, they found statistically significant trends, e.g. negative in East Europe and positive in the northern part of the North Atlantic. This pattern was linked with the opposite trends in the tropopause height. A continuation of the $TCO_3$ declining tendency since the ODS turnaround was surprisingly revealed in the NH lower stratosphere (Ball et al., 2018). Rather positive trend was expected due to a strengthening of the Brewer–Dobson circulation that was suggested by chemistry–climate models (e.g. Dietmüller et al. 2021). The negative trends in the NH lower stratosphere might be attributed to enhanced horizontal air mass exchange with the tropical region less abundant with ozone in the lower stratosphere (Thompson et al. 2021). However, other studies using observational data did not reveal such decline in the NH lower stratospheric ozone mostly due to large uncertainty in the satellite observations (e.g. Arosio et al. 2019).

NOAA proposed the Ozone Depleting Gas Index (ODGI) to keep track on changes in ODS concentration in the mid-latitudes and Antarctica (Montzka et al., 2022). Equivalent effective stratospheric chlorine (EESC) was used as a measure of the stratospheric halogen loading, which is weighted by the ozone destruction potential of each gas depleting stratospheric ozone. ODGI was defined as an indicator of the EESC current decline since its peak that is expressed in percentage of the corresponding decline needed to reach the EESC level in 1980, when ODS concentration in the stratosphere was only slightly affected by man-made substances. The EESC peak year in the mid-latitude stratosphere was found in 1997 being three years later than the ODS peak in the troposphere (Montzka et al., 2022). The corresponding ODGI value in 2020 was 51.7 % providing almost half reduction of EESC necessary to reach its undisturbed level existing in 1980. This also provided an estimate of the recovery time around 2045 in the mid-latitudes if factors affecting ODS changes were the same as those in the period 1997–2020. In addition, the EESC level in 2020 corresponds to the 1988 level (Fig. 1).

Following ODGI concept, we propose indices to monitor ozone recovery attributed to the EESC changes using various $TCO_3$ dataset from ground-based and satellite observations, and two reanalyzes. These indices will be calculated for selected NH mid-latitudinal sites corresponding to locations of the ground-based stations with long-term time series archived in the World Ozone and Ultraviolet Radiation Data Centre (WOUDC). From the smoothed pattern of long-term $TCO_3$ changes, we will extract $TCO_3$ values in key years: 1980, 1988, 1997 and 2020 (Fig. 1). These four values will be used to calculate the proposed ozone recovery indices (ORIs) showing the current stage of the stratospheric ozone recovery. The Goddard automailer (http://acdb-ext.gsfc.nasa.gov/Data_services/automailer/index.html) will be implemented to simulate various possible EESC patterns to estimate variability of the EESC peak year and the year when the EESC value is equal to its value in 2020.

In Sect. 2, the $TCO_3$ datasets are presented. In Sect. 3, the data preparation, ORI definition, and a method to obtain the uncertainty range of the ORI estimate are described. Section 4 presents ORI values for selected mid-latitudinal sites for various datasets using combinations of the data smoother and data category. In Sect. 5, the discussion and conclusions are presented.

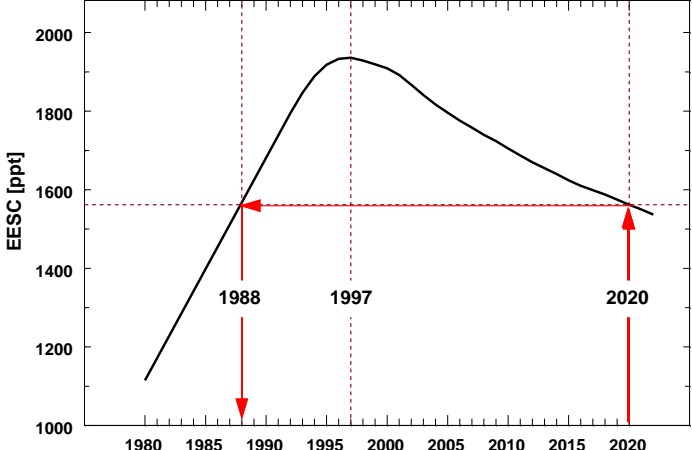

**Fig. 1. The EESC time series with marked key years: the EESC maximum in 1997 and 1988 when the EESC value was the same as in 2020 (the end of total column ozone data used in the paper) based on the EESC pattern proposed by Montzka et al. (2022).**

## 2 Total column ozone datasets

Four TCO$_3$ data sets are used in the study based on the ground-based and satellite observations as well as from two reanalyzes. The TCO$_3$ time series are calculated for selected sites in the NH mid-latitudes (35–60° N) corresponding to locations of stations that archived results of the ground-based observations by the Dobson and/or Brewer spectrophotometer at WOUDC. Arosa and Davos data sets were combined as the ozone monitoring at Arosa was moved to nearby station Davos in 2014. 16 stations with long-term and continuous observations (starting at least before 1980 and ending after 2020) are selected (Table 1). The monthly mean TCO$_3$ values for these stations are taken from the WOUDC web site.

**Table 1. Selected ground-based total column ozone observing stations in the NH mid-latitudes with the data record archived in WOUDC.**

| No. | WOUDC No. | Station | Lat. | Lon. | Elevation |
|---|---|---|---|---|---|
| | | Japan | | | |
| 1 | 14 | Tateno | 36.0°N | 140.1°E | 31 m |
| 2 | 12 | Sapporo | 43.1°N | 141.3°E | 19 m |
| | | North America | | | |
| 3 | 106 | Nashville | 36.3°N | 86.6°W | 182 m |
| 4 | 67 | Boulder | 40.1°N | 105.3°W | 1689 m |
| 5 | 65 | Toronto | 43.8°N | 79.5°W | 198 m |
| 6 | 19 | Bismarck | 46.8°N | 100.1°W | 511 m |
| 7 | 76 | Goose Bay | 53.3°N | 60.4°W | 40 m |
| 8 | 21 | Edmonton | 53.6°N | 114.1°W | 766 m |
| 9 | 77 | Churchill | 58.8°N | 94.1°W | 35 m |
| | | Europe | | | |
| 10 | 35 | Arosa | 46.8°N | 9.7°E | 1840 m |
| 11 | 99 | Hohenpeissenberg | 47.8°N | 11.0°E | 975 m |
| 12 | 96 | Hradec Kralove | 50.2°N | 15.9°E | 285 m |
| 13 | 53 | Uccle | 50.8°N | 4.4°E | 100 m |
| 14 | 68 | Belsk | 51.8°N | 20.8°E | 180 m |
| 15 | 165 | Oslo | 59.9°N | 10.7°E | 90 m |
| 16 | 43 | Lerwick | 60.0°N | 1.2°W | 80 m |

The NASA Merged Ozone Data (MOD) version 8.7 is used for a comparison with the WOUDC data. Overpass subset of MOD provides daily means of ozone content in various stratospheric layers and column ozone over the WOUDC sites including also those listed in Table 1. MOD time series are build using the homogenized spectral measurements of the solar

backscattered UV on various satellite platforms Nimbus 4, Nimbus 7, NOAA 9, 11, 14, 16, and 17–19, and Suomi National Polar-orbiting Partnership (NPP) (Frith et al., 2014; Weber et al., 2022).

Other two data sets represent the category of reanalyzed data, i.e., global $TCO_3$ field taken from output of 3-D chemistry and transport model. Here we examine time series for the selected sites interpolated from the Multi-Sensor Reanalysis, version 2 (MSR2) global data with the 0.5° x 0.5° resolution (van der A et al., 2015) and Modern-Era Retrospective analysis for Research and Applications, version 2 (MERRA2) with the 0.5° (latitude)x 0.625°(longitude) resolution (Wargan et al., 2017). Table 2 present the sources of all data sets used in the study.

**Table 2. Source of total column ozone datasets.**

| Datasets | Time Resolution | Data Start | WEB Page |
|---|---|---|---|
| WOUDC | Day | 1926 (Arosa) | https://woudc.org/archive/Summaries/TotalOzone/Daily_Summary/ |
| MOD | Day | 1970 | https://acd-ext.gsfc.nasa.gov/anonftp/toms/sbuv/MERGED/ |
| MSR2 | Month | 1979 | http://climexp.knmi.nl/select.cgi?field=o3col |
| MERRA2 | Month | 1980 | https://disc.gsfc.nasa.gov/datasets/M2TMNXCHM_5.12.4/summary |

## 3 Method

The $TCO_3$ monthly means from all data sets described in Sect. 2 are averaged over warm sub-period of the year (April–September) to build seasonal $TCO_3$ time series (1980–2020) for all NH mid-latitudinal selected sites listed in Table 1. The dynamic variability of ozone in these months is much smaller than in the cold-subperiod of the year, so the variability of $TCO_3$ is lower and chemical changes are potentially more pronounced. Moreover, ground-based $TCO_3$ observations are the most accurate in this part of the year due to the Sun's high elevations and better weather conditions, allowing the most reliable ozone observations with spectrophotometers (Komhyr, 1980).

During the warm sub-period of the year, UVR is much higher comparing to the cold sub-period. UV overexposures having detrimental effect on the Earth ecosystems are most frequent in the warm sub-period of the year. Therefore, analysis of the long-term $TCO_3$ changes in this period of the year is of special importance to discuss detrimental biological effects of the ozone changes. In the mid-1980s, the anticipated risks of UV overexposure led to international initiatives to protect the ozone layer. In addition, the results based on for the entire year (January–December) and the cold sub-period (October–next year March) data are presented in Appendix A.

### 3.1 Data smoother

Two data smoother types are examined in the study: LOWESS by Cleveland (1979) and super smoother (SS) by Friedman (1984). Here, for LOWESS application, the smoothness level is pre-defined to have up to two turning points in the smoothed time series, i.e., the smoothing parameter $f$ is set equal to 0.5. The first turning point corresponds to the ODS turning point in 1997 and next point is to reveal a change of the recovery rate after 1997 (if it exists). For SS, we use an option with not pre-defined smoothness level. The smoothed curve is obtained by means of a bivariate regression smoother based on local linear regression with adaptive bandwidths. Figure 2 illustrates the performance of both smoothers applied to the time series of seasonal (April–September) $TCO_3$ means for Tateno and Toronto from the ground-based observations.

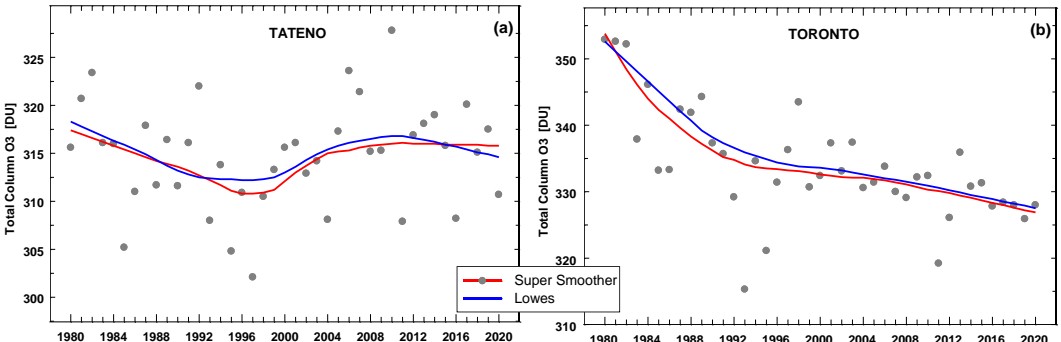

**Fig. 2. Examples of the long-term time series of total column ozone from ground-based observations in the warm sub-period (April–September) by locally weighted scatterplot smoother (blue curve) and super smoother (red curve): Tateno (a), Toronto (b).**

### 3.2 Removal of the ozone natural variability

The smoothed pattern of $TCO_3$ time series is used to discuss long-term variability in the series comprising 41 yearly values. The smoothers described in Sect. 3.1 are applied to both original time series and the series with removed variability due to various dynamical/chemical processes not directly involved with the anthropogenic emission of chemicals affecting the stratospheric ozone. Various proxies (explanatory variables in MLR regressions) for these "natural" processes have been proposed parameterizing dynamical/chemical forcing on the ozone layer. Frequently in MLR, the solar activity cycle (e.g. 10.7 cm solar radio flux), indices of internal atmospheric fluctuations (Quasi Biennial Oscillations, El-Niño Southern Oscillations, and Arctic Oscillations), optical depth of the stratospheric aerosols, and the eddy heat flux in the stratosphere (to parameterize the intensity in the Brewer–Dobson circulation) were used. The proxy set proposed by Weber et al. (2022) is used here with the sources listed in their Table 2. The only differences between our and Weber et. (2022) proxy set is using the eddy heat flux at 100 hPa from the NASA Atmospheric Chemistry and Dynamics Laboratory data base (https://acd-ext.gsfc.nasa.gov/Data_services/met/ann_data.html) and the stratospheric aerosol optical depth updated from Sato et al. (1993) for the entire time data (i.e., no different aerosol data sources after 1990).

The standard MLR with two independent linear trend terms (before and after the year of ODS turnaround) is used to extract the trend pattern and the combined signal due to all proxies. The MLR used here is identical to the one labeled as full MLR in Weber et. (2022). Finally, the $TCO_3$ time series combining proxy signals is subtracted from the original time series to obtain the $TCO_3$ series containing only the anthropogenic long-term component and noise. Further in the text, this time series is called non-proxy times series.

### 3.3 Ozone Recovery Index

The proposed ozone recovery indices (ORIs) follow the ODGI concept (Montzka et al., 2022) of using selected values from the EESC series to define the recovery status. Key $TCO_3$ values are taken at the beginning (1980), at the end of the data (2020), in the year of $TCO_3$ trend overturning (1997), and in the year (1988) when the EESC value in the NH mid-latitudes was the same as in 2020. Last two years are taken according the EESC pattern shown in Fig.1. Smoothed $TCO_3$ values in the selected key years $T$, denoted as $< TCO_3(T) >$, $T=\{1980, 1988, 1997, 2020\}$, are used for calculation of the following dimensionless ORIs in 2020:

$$ORI_1(1988, 2020) = 100\,\% \frac{< TCO_3(2020) > - < TCO_3(1988) >}{< TCO_3(1980) >} \tag{1}$$

$$ORI_2(1997, 2020) = 100\,\% \frac{< TCO_3(2020) > - < TCO_3(1997) >}{< TCO_3(1980) > - < TCO_3(1997) >} \tag{2}$$

If the ozone recovery follows ODS changes, $ORI_1(1988, 2020)$ will be equal to 0 % but negative (positive) for slower (faster) TCO$_3$ recovery than that found in the EESC pattern. Correspondingly, $ORI_2(1997, 2020)$ value in the Northern Hemisphere will be equal to 48.3 % (i.e. 100 % – ODGI(2020), see ODGI(2020)= 51.7 % in Montzka et al. 2022) if the ozone recovery in the period 1980-2020 follows EESC changes and lower (higher) than this reference value if the ozone recovery is slower (faster) comparing to that existing in the EESC. The ozone recovery will happen in the year TR when $<TCO_3(TR)>=<TCO_3(1980)>$. This means that the ozone recovered to its initial level when the stratosphere was not contaminated by man-made compounds (in 1980). In this case, $ORI_2(1997, TR)=100$ %.

If $ORI_2(1997, 2020)$ is less than 0 % and the ozone was declining before the ODS overturning year (usual case for the NH mid-latitudinal TCO$_3$), the ozone depletion will continue also after the EESC overturning. It is worth mentioning that for a searching of the TCO$_3$ recovery over Antarctica in 2020, the smoothed TCO$_3$ values in 1980, 1993, 2001, and 2020 need to be selected as these years correspond with key years in the EESC pattern for this region (Montzka et al., 2022).

## 3.4 ORI reference range

The specific EESC pattern, which is shown in Fig.1, was used in the calculation of ODGI and the year when EESC was equal its value in 2020. Various shapes of the EESC pattern can be provided via the Goddard automailer depending on parameters characterizing ODS in the stratosphere: mean age of air, age of air spectrum width, Bromine scaling factor, and the fractional release type. For mid-latitudes, the default values in the automailer are 3 yr and 60 for the mean age of the air and the Bromine scaling factor, respectively. We examine additional two options, i.e., 3.3 yr & 2.7 yr and 45 & 75, based on the uncertainties (1σ) of these parameters discussed by Velders and Daniel (2014) for mid-latitudes (see their Table 2). Moreover, two options are assumed for age of air spectrum width, i.e., 1.5 yr (default in the automailer) and 3 yr (arbitrarily selected). Moreover, three types of the fractional release type are implemented: Newman et al. (2006), Laube et al. (2010), and default type denoted as WMO_2010. In total, 54 EESC patterns have been obtained. Figure 3a shows the EESC's peak values and the corresponding years at the EESC maxima derived from the automailer together with the pair (1936 ppt and 1997) used in ODGI calculations by Montzka et al. (2022). The range of 100 % – ODGI(2020) values (equivalent to $ORI_2(1997, 2020)$ obtained with the corresponding EESC values instead of TCO$_3$ in Eq. (2) ) is from 40 % to 60 % using all simulated EESC curves (Fig. 3b). This gives the reference range for $ORI_2(1997, 2020)$ when the ozone recovery is similar to that of the EESC.

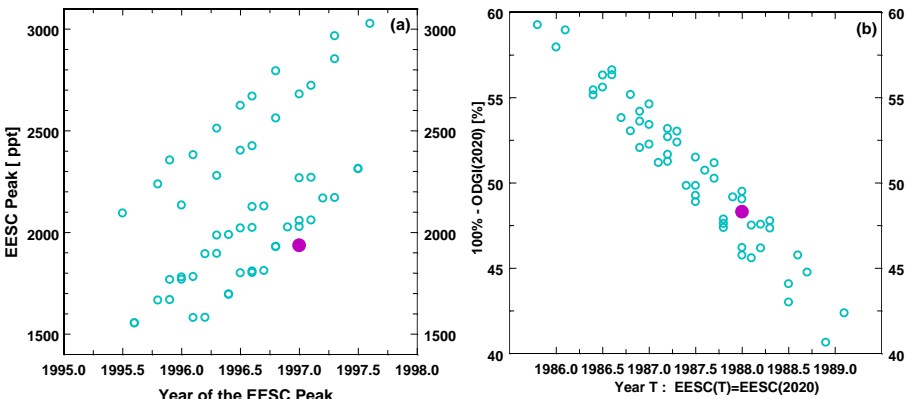

**Fig. 3. The EESC characteristics derived via the Goddard automailer (open circles) and those (full circles) taken from Montzka et al. (2022): EESC maximum value versus the year at the peak (a), $ORI_2$(1997, 2020) value calculated on the basis of the EESC values in Eq. (2) (equivalent to 100 %–ODGI(2020)) versus the year when EESC value is equal to that in 2020 (b).**

The year (before the EESC peak) $T$ when EESC was equal to its value in 2020 changes from 1986 to 1989 (Fig. 3b). The mean total ozone value, $< TCO_3(T) >$, for this year can be calculated assuming the ozone linear trend with rate $A$ (in Dobson Unit per year) as almost linear depletion of TCO$_3$ existed in mid-latitudes in the 1980s (e.g. Hudson et al., 2006):

$$< TCO_3(T) >= < TCO_3(1988) > + A(T - 1988) \tag{3}$$

For the earliest (1986) and latest year (1989) shown in Fig. 3b using the definition of $T$, $ORI_1 (T, 2020) \overset{\text{def}}{=} 0$, we obtain after simple mathematical manipulations with Eq. (1) and Eq. (3) that $ORI_1(1988, 2020)$ equals to $-2 \, \text{yr} \, A'$ and $1 \, \text{yr} \, A'$, respectively, when $A' = 100 \, \% \, A < TCO_3(1980) >^{-1}$ is the ozone trend (in percentage) in the 1980s. Earlier studies estimated $A'$ of a few percent per decade in the NH mid-latitudes (e.g. WMO, 1999; Hudson et al., 2006). Finally, taking $A'$

of –0.5 % per year, the reference range for $ORI_1(1988, 2020)$ can be estimated from –0.5 % up to 1 %.

## 3.5 Uncertainty of ORI estimates

The original TCO$_3$ time series, $TCO_3(t)$ is divided into two parts to account for the long-term variability, $< TCO_3(t) >$, which is extracted by the smoother, and the residual parts, $Resid\_TCO_3(t)$:

$$TCO_3(t) = < TCO_3(t) > + Resid\_TCO_3(t) \tag{4}$$

For each selected WOUDC station and data type (original or without proxy effects), two smoothers (LOWESS and SS) are applied to extract the long-term variability and the residual part of TCO$_3$ variability comprising dynamical/chemical part of variations and noise. The uncertainty range (between the 5th and the 95th percentile) of each ORI estimate is calculated from the set of ordered (lowest to highest) synthetic ORI values derived by bootstrapping. This method of calculations uncertainties of the trend estimates has been applied in our previous studies (e.g. Krzyścin et al., 2015).

First step of bootstrapping is building synthetic $n$-th time series, $Resid\_TCO_3^n(t)$, to mimic the residual term in Eq. (4),

$$Resid\_TCO_3^n(t) = Resid\_TCO_3(t_x), \qquad t_x \in \{ 1980, \dots, 2020\}, \quad n = \{1, \dots, N\} \tag{5}$$

Random year $t_x$ is obtained by drawing with replacement any year between 1980 and 2020. The total number of the bootstrapped time series used, $N$=10000, is derived experimentally to have stable estimates of ORIs. The use of the drawing with replacement in the construction of synthetic residuals was allowed because the original residuals were random, which

was confirmed by the Wald-Wolfowitz one sample run test (Wald and Wolfowitz, 1940).

The original and synthetic residuals should be identical in nature, i.e., have almost similar cumulative distribution. The Wald-Wolfowitz two samples run test is applied to check the difference between the original and synthetic residuals. If the time series of residuals derived by the drawing with replacement was found to be significantly different (with the 95 % confidence) from the original ones, such the residuals were not used in the bootstrapping. This means that the number of draws

was greater than the total number of the series ($N$) used in the bootstrap sample.

Next step of the bootstrapping is adding $Resid\_TCO_3^n$ term to the smoothed part of the original TCO$_3$ series to build $n$-th synthetic $TCO_3(t)$ time series:

$$TCO_3^n(t) = < TCO_3(t) > + Resid\_TCO_3^n(t) \tag{6}$$

The smoothed $n$-th time series, $< TCO_3^n(t) >$ is obtained applying LOWESS (or SS) to synthetic $TCO_3^n(t)$ time series.

Repeating this procedure, a set of synthetic series, $TCO_3^n(t)$, $n = \{1, \dots N\}$, is constructed.

Finally, for each $< TCO_3^n(t) >$ series, $ORI_1^n(1988, 2020)$ and $ORI_2^n(1997, 2020)$ are calculated using Eq. (1) and Eq. (2), respectively. From the ordered set of $ORI_1^n(1988, 2020)$ and $ORI_2^n(1997, 2020)$ values, the uncertainty range (5th percentile–95th percentile) is obtained. This allows to discuss if ORIs values are significantly different than the reference ORI ranges (Sect. 3.4) defining the recovery stage, i.e., –0.5 % to 1.0 % and 40 % to 60 % for $ORI_1(1988, 2020)$ and

$ORI_2(1997, 2020)$, respectively.

## 4 Results

Figures 4–5 illustrate the median and uncertainty range of $ORI_1(1988, 2020)$ and $ORI_2(1997, 2020)$, respectively, for all WOUDC stations listed in Table 1. These values are calculated using $TCO_3$ monthly mean values averaged over the warm sub-period of the year. Results are shown in Figs. 4–5 (a–d) for WOUDC, MOD, MSR2, and MERRA2 data, respectively.
The order of the stations on "x" axis in these Figures is the same as in Table 1, i.e., first there are two Japanese stations, then seven North American, and finally seven European stations. For each region (between the vertical dashed lines), the results are arranged from the southernmost to the northernmost station.

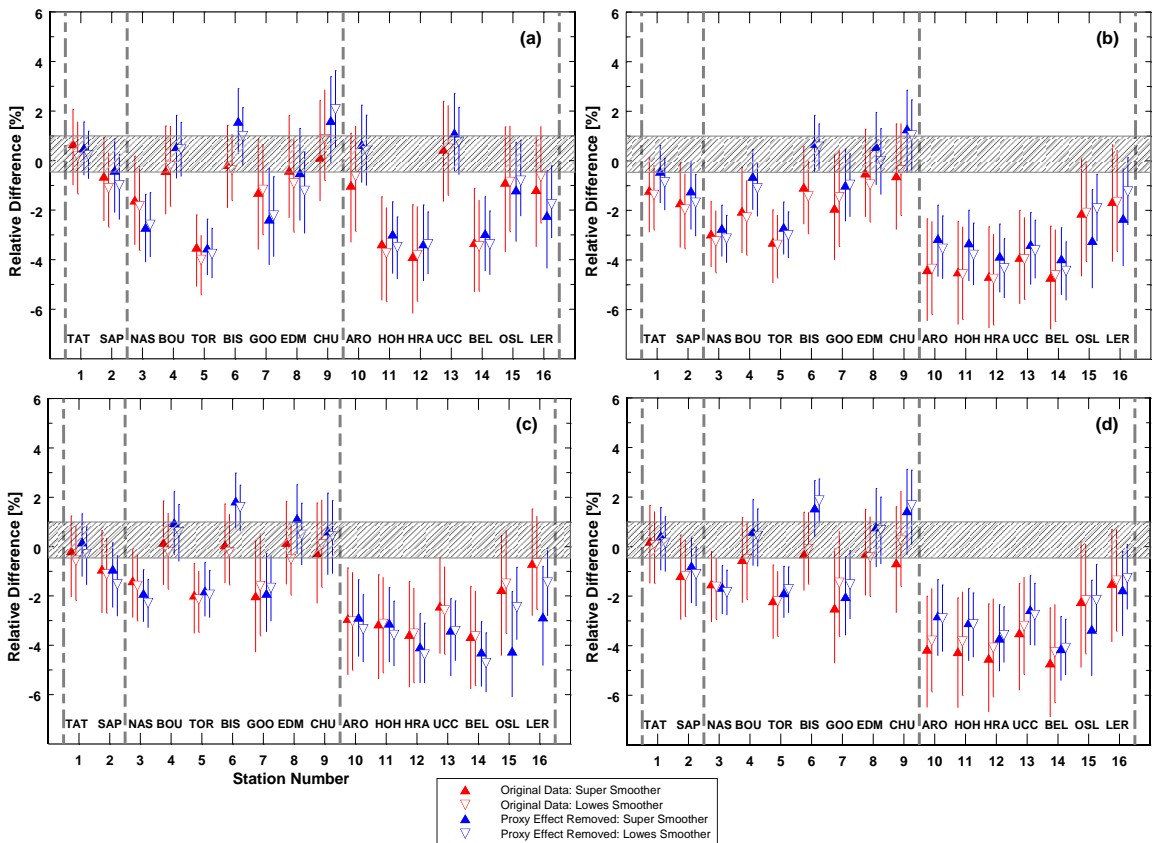

**Fig. 4. Median of the ozone recovery index, $ORI_1(1988, 2020)$, estimate (triangle) based on Eq. (1) and its uncertainty range (line) for**
**16 NH mid-latitudinal stations (Table 1) and various combination of data smoothers and datasets: WOUDC data (a), MOD data (b), MSR2 (c), and MERRA2 (d). Results in red are for the original data and in blue for the data with removed natural variability. The hatched area marks the reference range, –0.5 % to 1.0 %, calculated in Sect. 3.4.**

The location of the uncertainty range of ORIs in Figs. 4–5 relative to its reference range defined in Sect. 3.4 provides the stage of the ozone recovery at each selected station. Namely, if the uncertainty range of ORI is entirely above (below) the
245 reference range, the ozone recovery is faster (slower) than that existing in the EESC for the period 1980–2020. This means that the $TCO_3$ recovery rate is significantly different (at least at the 95 % confidence level) when compared with corresponding the EESC values. If the uncertainty range crosses the reference range, the hypothesis of the $TCO_3$ recovery rate following the EESC changes cannot be rejected. The depletion of $TCO_3$ will continue even after the EESC maximum if the uncertainty range for $ORI_2(1997, 2020)$ is below 0 %. In this case, there is no $TCO_3$ recovery for the entire period (1980–2020).
The ozone recovery is discussed both for the original data and non-proxy time series (with removed combined proxy signal from the original data). A strong signal that the $TCO_3$ recovery at a selected station is slower (faster) than that found in the EESC if the uncertainty ranges of both ORIs are below (above) the reference ranges for all data types used (WOUDC, MOD, MSR2, and MERRA2), regardless of the smoother type used.

Strong support for the slower TCO$_3$ recovery can be found for three European stations: Hohenpeissenberg, Hradec Kralove, and Belsk. For Arosa and Uccle, using $ORI_2(1997, 2020)$, the slower ozone recovery is found in all datasets except WOUDC. This may suggest a problem with the homogeneity of WOUDC data at least for Arosa, as the results are different than those obtained at the nearby Hohenpeissenberg station. For Toronto and Nashville, using $ORI_2(1997, 2020)$ derived from the non-proxy time series, the slower ozone recovery can be suggested.

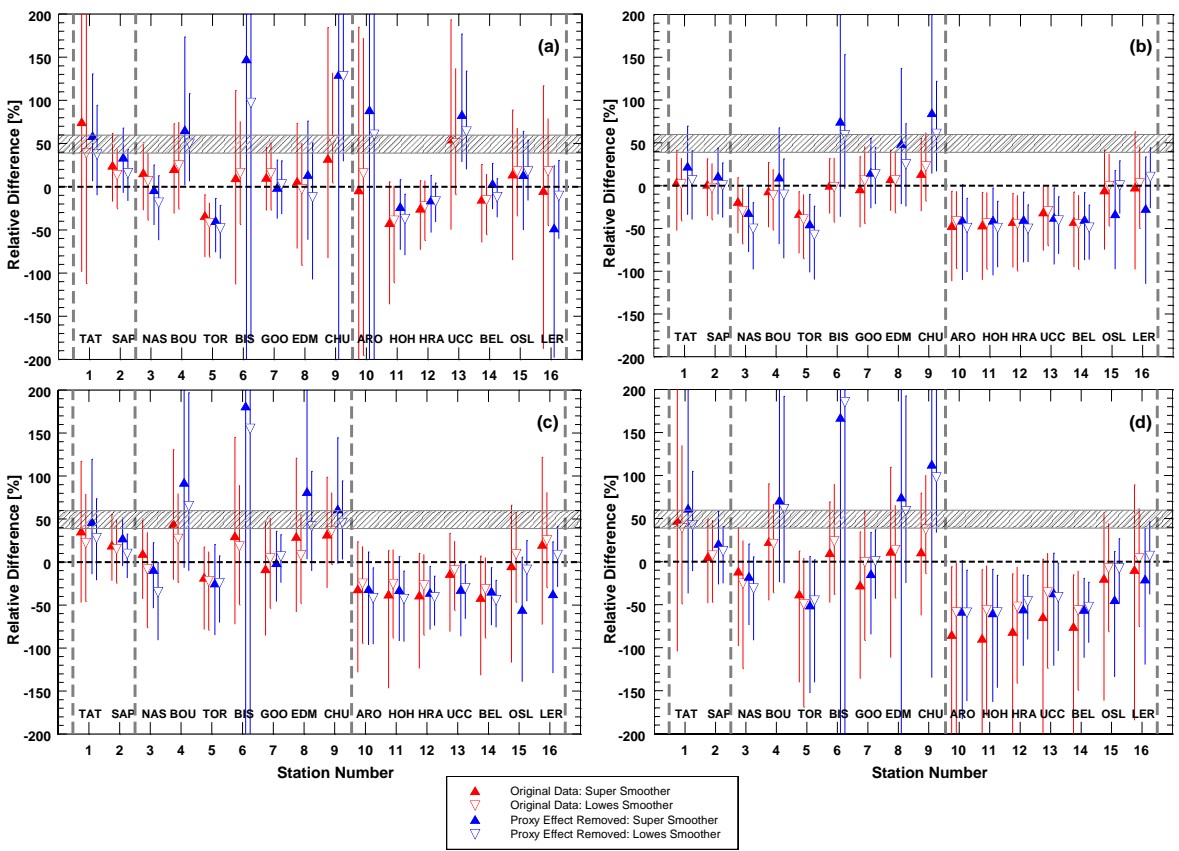

**Fig. 5. The same as Fig. 4 but for the ozone recovery index, *ORI₂*(1997, 2020), based on Eq. (2). The reference range (hatched area) is between 40 % and 60 % according estimates in Sect. 3.4.**

There are many individual cases where the localization of the uncertainty range of ORI estimate suggests significant difference between the TCO$_3$ and EESC recovery rates in at least one or two datasets. For example, this is the case for Nashville for both ORIs derived from MOD data (Fig. 4b and Fig. 5b). The differences between ORIs values derived from the original and non-proxy series sometimes appear, e.g. the non-proxy time series for Oslo shows slower recovery using $ORI_2(1997, 2020)$ for all data types excluding WOUDC but this is not found in the original time series.

A consistent pattern of the ORIs variability is found for all data records. Lower ORIs for Nashville and Toronto were calculated among the North American stations. Moreover, ORIs below the reference range appeared in Europe (except two northernmost stations). This was not found in Arosa and Uccle using-ground-based data probably due to instrumental problems. ORIs values indicated that the TCO$_3$ recovery in Japan follows the EESC change. For this region, slower recovery was found only in the MOD original data for Sapporo using $ORI_2(1997, 2020)$. Usually, uncertainty of ORIs estimates in MOD time series were the smallest that allows to identify additional sites with slower recovery (e.g. Nashville for the original and non-proxy time series, Fig.5b). In all cases, the TCO$_3$ recovery was never faster than that in the EESC pattern.

A continuation of ozone decline after the turnaround in ODS concentration, which appears when the uncertainty range of $ORI_2(1997, 2020)$ is entirely below 0 regardless of the smoother type applied, is found both in the original and non-proxy time series from WOUDC (Toronto, see also Fig. 2b for the TCO$_3$ time series), MOD (Toronto, Arosa, Hohenpeissenberg, Uccle, Hradec Kralove, and Belsk), and MERRA2 data (Arosa, Hohenpeissenberg, Hradec Kralove, and Belsk).

## 5 Discussion and conclusions

We proposed a novel tool to examine the stage of ozone recovery attributed to the EESC's change. We introduce two ORIs. The first ORI is a difference between the TCO$_3$ value in 2020 and that in the year before the EESC maximum with the same EESC value as that in 2020. The second ORI is the percentage of the recovered ozone since the EESC maximum. The following ozone recovery phases can be identified when compared with the EESC change, i.e., faster, slower, no conclusive (the hypothesis of the TCO$_3$ recovery driven entirely by EESC change cannot be rejected), and a continuation of TCO$_3$ decline after the EESC peak. For NH mid-latitudes in 2020, first three categories are found from the location of the ORI uncertainty ranges relative to the reference ranges, from –0.5 % to 1 % for $ORI_1(1988, 2020)$ or from 40 % to 60 % for $ORI_2(1997, 2020)$. The reference ranges were obtained from simulations of the mid-latitudinal EESC time series via the Goddard automailer. The last category appears when the uncertainty range of $ORI_2$ is completely below its 0 % reference line.

The stage of ozone recovery level was usually discussed comparing linear trend values before and after the year of the EESC overturning in the mid-1990s for the NH mid-latitudes. The trends were obtained using MLR applied to various TCO$_3$ datasets and the slopes of the trend lines, which can be joint (Reinsel et al., 2005) or disjoint (Weber et al., 2018) at the EESC turning year, were compared. Weber et al. (2022) found that for the increasing rate of near global ozone (60°S–60°N) after 1995 was roughly a third of the decreasing rate calculated in the period 1978–1995 that corresponds with the ratio of the EESC linear change after and before the EESC maximum. This supported the success of MP and its further amendments. For this case, corresponding $ORI_2(2020)$ of ~47 % (i.e., inside the reference range) is estimated from the ratio between the TCO$_3$ trends (1/3) multiplied by the ratio between duration of the period after (24 yr) and before (17 yr) the EESC maximum. This estimate is close to the corresponding 100 %–ODGI(2020) value equal to 48.3 % when ODGI(2020)=51.7 % is taken according to Montzka et al. (2022) calculations (see ODGI(2020) in their Table 2). Then, two approaches, which are based on the TCO$_3$ trends and $ORI_2(2020)$, are almost equivalent and provide the TCO$_3$ recovery corresponding with the EESC change.

In our approach, to disclose phase of the ozone recovery, we compare ORIs, which are derived from the smoothed pattern of the ozone time series, with corresponding indicators of the mid-latitude EESC recovery. ORIs are calculated for each selected mid-latitudinal WOUDC station with the long time TCO$_3$ series. 16 stations are selected and various datasets for these stations based on the ground-based measurements, satellite overpasses, and two reanalyzes, are examined. For almost all stations and data types, the ozone recovery follows EESC pattern. Only in few cases it is slower. The firm sign of the slower ozone recovery is identified for a group of neighborhood central/western Europe sites (Hohenpeissenberg, Hradec Kralove, and Belsk) as all ORI uncertainty ranges, regardless of the smoother and data type, are below the reference ORI ranges. This means that TCO$_3$ trends are less than expected from the EESC change. The smallest uncertainty ranges of ORIs estimates are for the MOD dataset that result in the larger number of sites with slower TCO$_3$ recovery.

A negative $ORI_2(1997, 2020)$ value means that the mean TCO$_3$ level in 2020 is lower than that in 1997 as the denominator in Eq. (2) is positive because TCO$_3$ was declining in NH mid-latitudes before the EESC overturning. When the uncertainty range of $ORI_2(1997, 2020)$ is completely below zero, this means that the decline in ozone continues even after the EESC peak. This case is found for the group of central/western Europe stations from MOD and MERRA2 data. The continuation of ozone depletion in this region was previously discussed by Coldewey-Egbers et al. (2022) using merged satellite data.

Negative trends in TCO$_3$ since the EESC peak provides lower TCO$_3$ values at the end of time series (2020) and also negative ORIs thereafter. Szelag et al. (2020) reveals negative trends in the ozone vertical profile ranging from −1 % per decade to −2 % per decade in the lower/middle stratosphere at 30–60° N for the period 2000–2018 during the summer (June−July−August). Moreover, according to WMO (2022), declining tendency in the tropospheric ozone due to air quality improvement in some regions can provide additional source for TCO$_3$ depletion for sites where the troposphere was cleaned from the ozone precursors. This is probable for the central/western European sites with $ORI_2(1997, 2020)<0$ %. Superposition of the negative

ozone trend in the lower/middle stratosphere over NH mid-latitudes in summer and the troposphere cleaning supports possibility of negative $ORI_2(1997, 2020)$ value that also means no ozone recovery for the entire period (1980–2020).

Recent studies have focused on anthropogenic trends extracted by MLR to assess changes in the ozone layer by halogens, which implies parameterization of both trend and the natural ozone variability. In this study, two data types, the original and with the combined proxy effects removed, are under interests. The former is used to quantify changes in the UV radiation reaching the Earth's surface caused by ozone and the latter one is to delineate anthropogenic changes (due to man-made halogens) in the ozone layer. In the main text, we examine the $TCO_3$ means for the warm period of the year, since UVR levels are naturally high during this period. If excessive UV exposure occurs due to lower $TCO_3$ values, it will cause serious consequences for health and the environment (Barnes et al., 2019). ORIs from the original and non-proxy datasets can sometimes be different, e.g. $ORI_2(1997, 2020)$ based on the non-proxy MSR2 data for Belsk and Hradec Kralove shows a continuation of ozone decline, but the slower recovery stage only comes from the original data (Fig. 5c).

Similar analysis for the monthly mean $TCO_3$ data averaged of the entire year (January–December, Figs. A1–A2) and the cold sub-period of the year (October–next year March, , Figs. A3–A4) are shown in Appendix A. For only a few cases, a faster $TCO_3$ recovery rate in the period 1980–2020 than that in the EESC series is found using the ground-based data for the cold sub-period of the year. This was identified in Arosa, Uccle, and Oslo when using $ORI_1(1988, 2020)$ after removal proxy effects from the original data (Fig. A3a). In Oslo, the faster $TCO_3$ recovery is also found in the original WOUDC data. Uccle is only one station where $ORI_2(1997, 2020)$ is above the reference range using the non-proxy $TCO_3$ time series (Fig. A4a) that suggests a faster $TCO_3$ recovery. These results for the cold-sub-period of the year should be treated with caution as the ground-based $TCO_3$ observations in this part of the year are usually less precise than those taken in the warm sub-period because of weather conditions (many cloudy days) and low solar elevation near (Komhyr, 1980). In the cold sub-period, slower $TCO_3$ recovery than that in the EESC time series was found only in Sapporo (Fig. A4b and Fig. A4d). Using the yearly mean $TCO_3$, slower recovery was revealed at fewer sites compared to data for the warm sub-period. In addition, this was also identified in Sapporo (Fig. A2b and Fig. A2d). However, a continuation of the ozone decline after the EESC peak for the European stations cannot be revealed. The differences between the European stations and other stations were not as pronounced as in the case of data from the warm-subperiod.

The variability of the ozone layer caused by the gradual removal of long-lived halogens from the stratosphere and changes in dynamic processes in the atmosphere, which are to some extent related to climate change, continue to pose a threat to the environment and health. The ozone recovery could be delayed (if it ever happens) in some isolated areas. The ozone issue, which was raised in the early 1980s due to the anticipated UVR increase is still worth considering. This study is a kind of an introduction to use ORIs, which are based on the smoothed $TCO_3$ values in key EESC years, to monitor the current state of the ozone layer and its relationship with changes in man-made halogen loading in the stratosphere. We plan to use the ORI concept to discuss the ozone recovery globally from various gridded datasets.

**Appendix A**

These supporting figures provide further insight into ORIs performances for 16 mid-latitudinal stations using averaged monthly mean $TCO_3$ values over the entire year (January–December) and cold sub-period of the year (October– next year March). The same data and smoother type were examined as those for ORIs based on the $TCO_3$ values averaged over the warm sub-period of the year (Figs. 4–5).

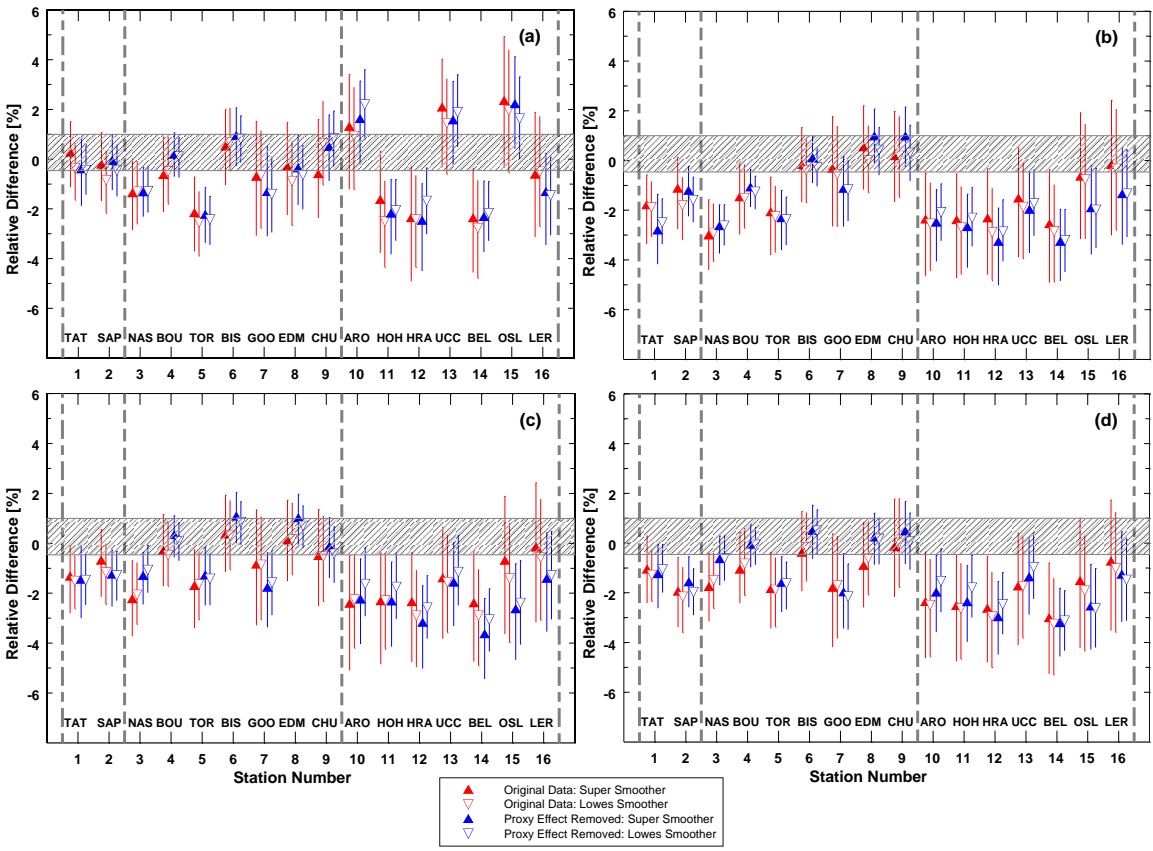

**Fig. A1. The same as Fig. 4 but for the ozone recovery index, *ORI*₁(1997, 2020), based on the TCO₃ monthly means averaged over the entire year (January--December).**

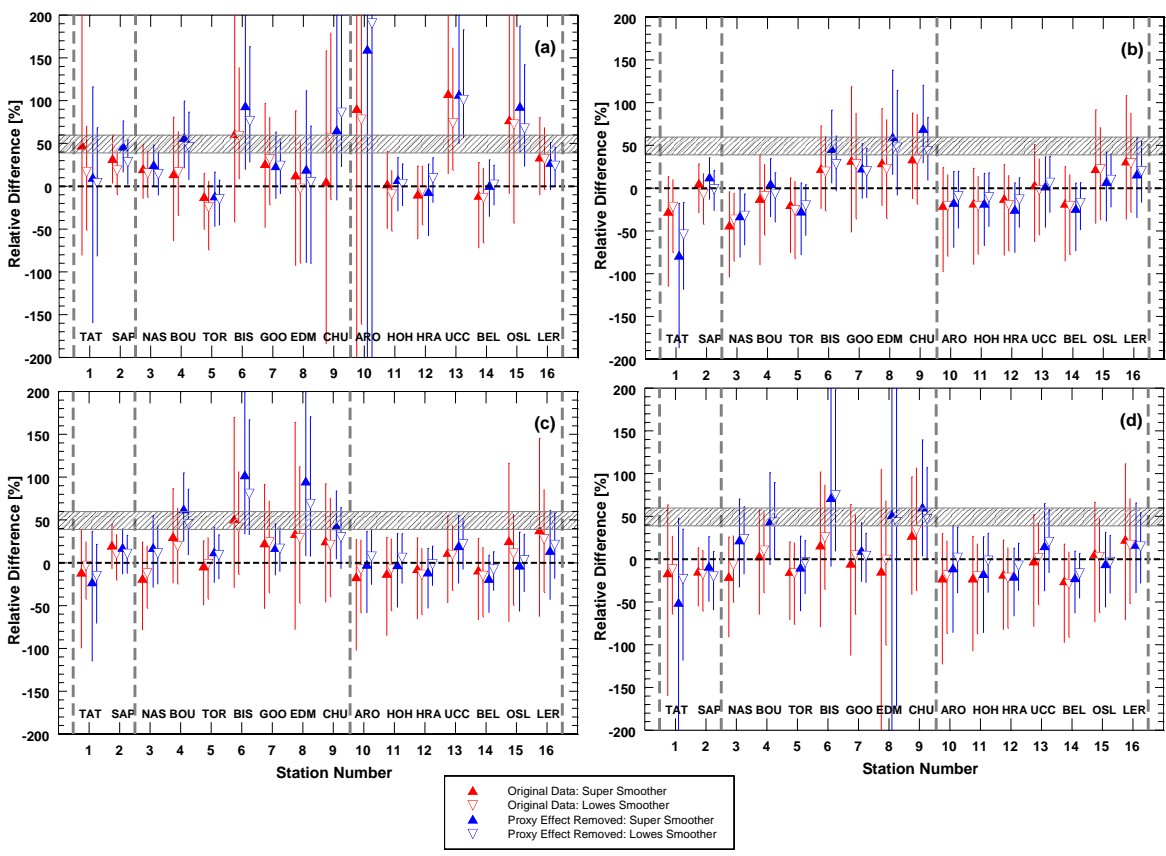

**Fig. A2. The same as Fig. A1 but for the ozone recovery index, *ORI*₂(1997, 2020).**

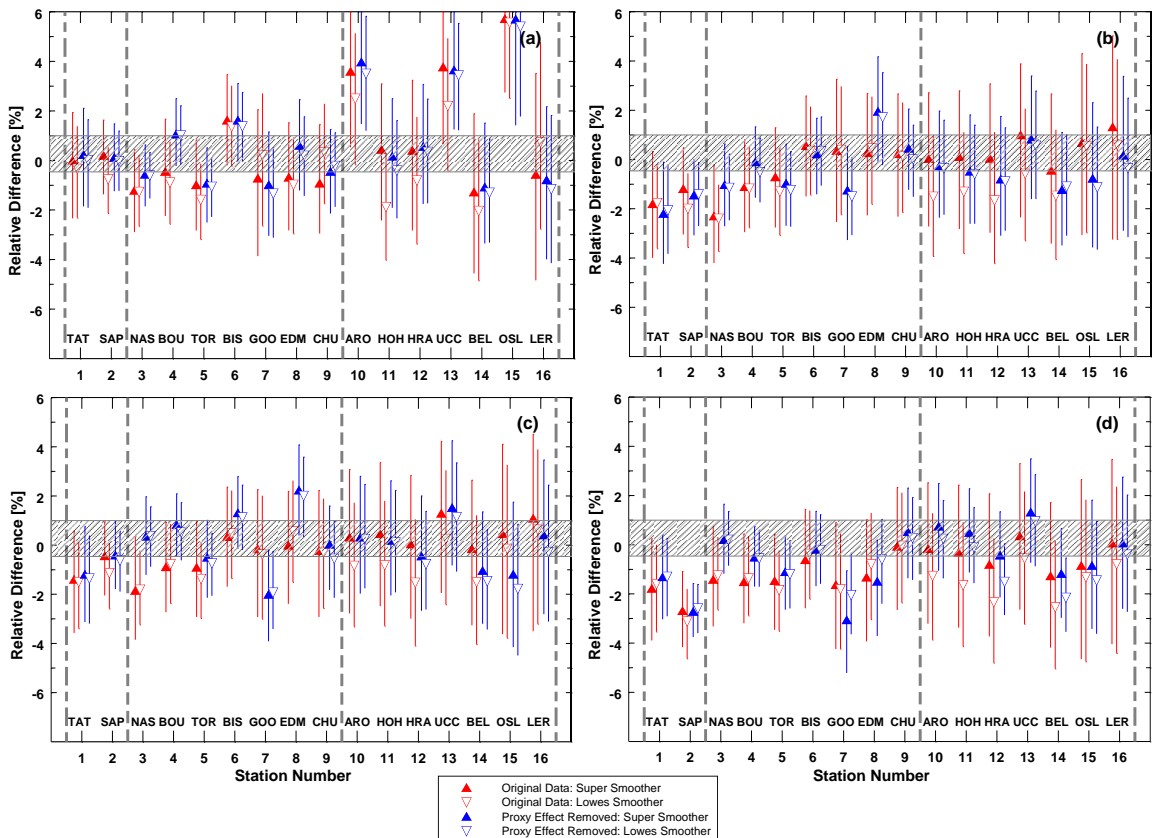

**Fig. A3** The same as Fig. A1 but for *ORI₁*(1997, 2020) based on the TCO₃ monthly means averaged over the cold sub-period of the year (October—next year March).

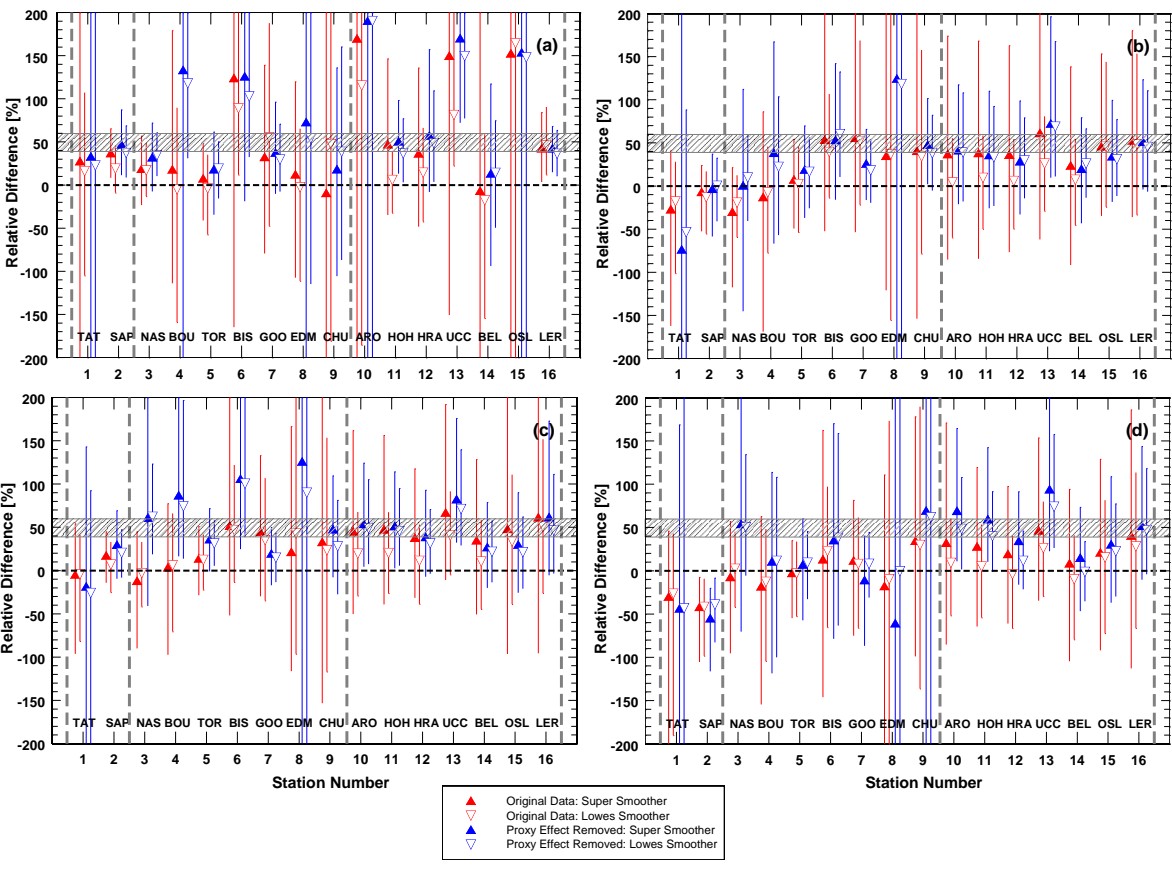

**Fig. A4. The same as Fig. A3 but for the ozone recovery index, *ORI₂*(1997, 2020).**

*Data availability*. The TCO$_3$ data used in this study are taken from the WEB sources, listed in Table 2, with non-limited access. "Full model" version of MLR by Weber at al. (2022) is applied to remove "natural" variability form the original TCO$_3$ data. The MLR proxy set is taken from the WEB sources listed in their Table 2.

*Author contributions*. JK contributed to all aspects of the data analysis, manuscript preparation, and interpretation of the results.

*Competing interests*. The author declares that he has no conflict of interest.

*Acknowledgments*. We thank the Chief Inspectorate of Environmental Protection, Poland, for funding the ozone measurements at Belsk.

*Funding*. This work was supported by the Ministry of Science and Higher Education of Poland under Grant number 3841/E-41/S/2022.

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
