# Peer review of "Indicators of the ozone recovery for selected sites in the Northern Hemisphere mid-latitudes derived from various total column ozone datasets (1980–2020)"

_Atmospheric Chemistry and Physics, 2022_

## Author Response (AR1)

**Reviewer 1**

**Original reviewer's remarks are in italic.**

**"Overall remarks:"**

Major problem No.1

"The other ozone recovery indicator ORI2(T), compares ozone recovery from 1997 to T with the expected maximum ozone depletion, from 1980 to 1997. ORI2(T) would be 100% for recovery to 1980s ozone levele, and would be 0% in 1997. This seems to be inconsistent with the EESC / ODGI plot in Fig. 1, where ODGI is at 100% in 1997, at 51.7% in 2020, and at 0% in 1980. I therefore believe that the expected value for ORI2(2020) should be 48.3% (=100%-51.7%), not 51.7%. Both values are close to 50%, though, and this difference / inconsistency is minor. It should however be checked carefully and (if necessary) fixed throughout the paper."

In the revised paper we correct this error. We explain that ORI2 corresponds with 1-ODGI and 48.3% is used throughout the manuscript instead of previous wrong number of 51.7%: "…Correspondingly, $ORI_2$(1997, 2020) value in the Northern Hemisphere will be equal to 48.3% (i.e. 100% - ODGI(2020), see ODGI(2020)= 51.7% in Montzka et al. 2022) if the ozone recovery in the period 1980-2020 follows EESC changes and lower (higher) than this reference value if the ozone recovery is slower (faster) comparing to that existing in the EESC. The ozone recovery will happen in the year TR when $<TCO_3 (TR)> = <TCO_3 (1980)>$. This means that the ozone recovered to its initial level when the stratosphere was not contaminated by man-made compounds (in 1980). In this case, $ORI_2$(1997, 2020)=100 %." P.6, l.158-163

Major problem No.2

"A second and larger problem, to me, is not mentioning the considerable uncertainty associated with using one EESC / ODGI time series only. Depending on the assumed age of air distribution, bromine contribution, data sources, ... there is a considerable spread between various EESC time series, see e.g. Newman et al. (2007), Engel et al. (2018), or the various WMO/UNEP Ozone Assessments. Thus, I think it is absolutely essential to discuss the substantial uncertainties associated with the ORI references values of 0% and 51.7% (or 48.3% ?). I think, Fig. 1 should show some uncertainty (e.g. by at least adding the Antarctic EESC /ODGI curve, see also my atttached Fig. 1). In Figs. 3 and 4 of the manuscript, the reference lines should really be reference ranges, indictae e.g. by much thicker lines, to reflect their uncertainty. Uncertainty of the EESCs and reference values, in my opinion, needs to be dicussed in nearly all the sections of the paper, because it is important for expected total ozone columns / reference values for the ORIs."

We use the Goddard automailer with different input parameters (54 cases were considered) to calculate the EESC variability and estimate uncertainty ranges for the ozone recovery index (ORI) references. IF ORI values are withing these ranges we can state that the ozone recovery follows EESC changes. New Sect. 3.4 has been added with new Fig. 3 summarizing results of EESC simulations by the automailer. These ranges are embedded to all figures showing ORIs calculations (Figs.4-5 and Fig.A1-A4). P.6/7, l.168-194 (Sect. 3.4).

"Overall, however, this is a good manuscript which presents useful new metrics for ozone layer recovery. It is well suited for ACP and should be published, after addressing my questions above, and the following minor suggestions."

**"Minor suggestions:"**

Lines 19 to 21: It would be good to add the respective years here: 1980, 1988, 1997, 2020.
These years have been added:
"These are key years in the equivalent effective stratospheric chlorine (EESC) time series for the period 1980-2020, i.e., the stratosphere was only slightly contaminated by ODS in 1980, 1988 is the year in which the EESC value is equal to its value at the end (2020), and the EESC maximum was in 1997 in mid-latitude stratosphere" P.1, l. 18-20.

"Line 23 and many other places: 51.7% or 48.3%? as mentioned above. Please check, throughout the paper."

The correct value of 48.3% replaced previous erroneous value of 51.7% in the revised manuscript.

"Line 61: Should be NOAA, not NASA."
NASA has been changed to NOAA in the revised text

"Line 61 to 70: This is one of the places where a dicussion of different EESC realizations and the accompanying uncertainties is necessary".
New Section (Sect. 3.4) has been added to show how the uncertainty ranges for ORIs have been calculated.

"Fig. 1, caption: Data source needs to be mentioned (https://gml.noaa.gov/odgi/ ?). Additional EESCs? Uncertainty?"
Revised caption provides the source of the data (Montzka et al., 2022)
"Fig. 1. The EESC time series with marked key years: the EESC maximum in 1997 and 1988 when the EESC value was the same as in 2020 (the end of total column ozone data used in the paper) based on the EESC pattern proposed by Montzka et al. (2022)." P.3, l. 83-84.
Uncertainty of the EESC pattern is discussed in new Sect. 3.4.

"Lines 105 to 110: Also mention that dynamical variability is much smaller in these months, so total ozone variability is smaller and chemical changes are potentially more clear."
This statement has been added:
"The dynamic variability of ozone in these months is much smaller than in the cold-subperiod of the year, so the variability of $TCO_3$ is lower and chemical changes are potentially more pronounced" P.4, l. 109-111.

"Figs. 3 and 4: As mentioned, show some uncertainty for the ORIs, e.g. make ORI reference lines wider, to account for EESC uncertainty."
It has been done, see new Fig.4-5 and Fig.A1-A4.

"The striking difference between the European stations and the other stations in Figs 3 and 4 should be discussed a bit more. It is mentioned very briefly in context with Coldewey-Egbers (2022) in the conclusions (around line 265), but it should be mentioned already when discussing the Figures. The paragraph in the conclusions should be expanded a bit as well."
New paragraph in Sect. 4 (Results) has been added to show differences between the regions and data types. Page 9, l. 261-267.

"The recent 2022 WMO/UNEP Ozone Assessment (https://csl.noaa.gov/assessments/ozone/) mentions tropospheric ozone changes as a potentially important factor in total column ozone changes in recent years. Improved air quality in Europe, for example, might have contributed to the lack of increases in total column ozone. I think it would be good to add some thoughts on this, e.g. an additional paragraph in the conclusions."
New paragraph in Sect. 5 has been added to discuss recent trends in total column ozone including results from WMO/UNEP Ozone Assessment. P.10, l. 299-316.

**Reviewer 2.**

**Original reviewer's remarks are in italic.**

**Overall remark**

*"…In my view the new indices provide a useful addition to the standard ozone trend analyses in order to assess the current stage of recovery."*

**General comments**

*"The author uses ozone values that were averaged over the warm sub-period of the year (Apr-Sep), which is justified with (i) higher solar elevation (leading to more accurate observations) and (ii) higher UV indices (having detrimental biological effects). However, regarding long-term ozone changes, some seasonal dependence might be expected (see, e.g., Szelag et al., 2020 or Coldewey-Egbers et al., 2022). I think it is necessary to investigate and discuss the uncertainty of the ORIs associated with ozone values averaged over different periods of the year, e.g., annual mean, cold sub-period (Oct-Mar), or seasonal means (DJF, MAM, JJA, SON). I suggest to add a figure indicating this uncertainty."*
New figures showing ORIs based on the ozone data averaged over the entire year and the cold subperiod of the year are shown in Appendix A. Results for these figures are discussed in added paragraph in Sect. 5. P. 11, l. 313-325.

*"Sec. 3: What is the impact of the Mt. Pinatubo eruption in 1991 and the extremely low ozone values in the following years on your results? Did you include these years in your analysis? Do all data records provide data during this time or do they have gaps?"*
The effects of the Mt. Pinatubo eruption on ozone are accounted for using aerosols proxy (stratospheric aerosol optical depth at 550 nm) in the multiple regression model for the case of the non-proxy time series (i.e., series with removed natural variability). The key-year for ORIs calculations, i.e., the year of the EESC maximum (1997) is far from the year of Mt. Pinatubo eruption (1991), so it has only slight effect on ORIs.

*"Sec. 4: The discussion of Figures 3 and 4 should be expanded a bit. I suggest to elaborate on the apparent difference between the regions (Europe and North America / Japan) and on the agreement/differences between the data records (WOUDC/MOD/MERRA-2/MSR2)."*
New paragraph has been added to discuss differences between regions and the data types. P.9, l.261-267.

*"Figs. 3 and 4: It might be helpful for the readers to somehow highlight those stations in the figures, which indicate a significant difference from EESC values."*
The uncertainty band (hatched area) has been added to Figures 4-5 showing range of ORI values supporting total column ozone recovery similar to that in the EESC pattern. In this case, stations with ORI showing faster or slower ozone recovery are more pronounced as for these stations the uncertainty lines of ORI (i.e., 5-95$^{th}$ percentile line) do not cross the hatched areas.

**Technical corrections**

P1, L14: „obtained" → „obtain"
P1, L35: „Earth surface" → „Earth's surface"
P1, L37: „Earth environment" → „Earth's environment"

P2, L51: „longitude/longitude" → „latitude/longitude"
P2, L61: „NASA" → „NOAA"
P3, L94: Please add also "Frith et al., 2014".
All these corrections have been included in the revised manuscript.

P5, Fig.2 caption: Which dataset is shown here? Ground-based? Pleas add the information in the caption.
The caption has been changed:
"Fig. 2. Examples of the long-term time series of total column ozone from ground-based observations in the warm sub-period (April–September) by locally weighted scatterplot smoother (blue curve) and super smoother (red curve): Tateno (a), Toronto (b)." P.5, l. 128-129.

P7, L207: „0%e" → „0%"
"e" has been deleted

P11, LL346-348: Paper now published in ACP: Maillard Barras, E., Haefele, A., Stübi, R., Jouberton, A., Schill, H., Petropavlovskikh, I., Miyagawa, K., Stanek, M., and Froidevaux, L.: Dynamical linear modeling estimates of long-term ozone trends from homogenized Dobson Umkehr profiles at Arosa/Davos, Switzerland, Atmos. Chem. Phys., 22, 14283–14302, https://doi.org/10.5194/acp-22-14283-2022, 2022.
Recently published paper by Maillard Barras et al. (2022) replaced their previous ACPD paper. This new paper has been added to references and older version deleted.

**References**

The suggested papers by Frith et al. (2014) and Szelag et al. (2020) are included into the manuscript.

---

## Author Response (AR2)

**Response to Referee #2,**

Referee #2 suggested to following technical corrections:

p.2, l. 57 add a comma: "(Thompson et al. 2021)" -> "(Thompson et al., 2021)"
p.2, l.59 add a comma: "(e.g. Arosio et al. 2019)" -> "(e.g. Arosio et al., 2019)"
p.12, Fig. A1 caption: "ORI1(1997,2020)" -> "ORI1(1988,2020)"
p.13, Fig. A3 caption: "ORI1(1997,2020)" -> "ORI1(1988,2020)"

**Author answer**:
All these corrections are included in the revised manuscript.